# Overcoming Lookback Window Limitations: Exploring Longer Windows in Long-Term Time Series Forecasting

## Abstract

Long-term time series forecasting (LTSF) aims to predict future trends based on historical data. While longer lookback windows theoretically provide more comprehensive insights, current Transformer-based models face the Lookback Window Limitation (LWL). On one hand, longer windows introduce redundant information, which can hinder model learning. On the other hand, Transformers tend to overfit temporal noise rather than extract meaningful temporal information when dealing with longer sequences, compounded by their quadratic complexity. In this paper, we aim to overcome LWL, enabling models to leverage more historical information for improved performance. Specifically, to mitigate information redundancy, we introduce the Information Bottleneck Filter (IBF), which applies information bottleneck theory to extract essential subsequences from the input. Additionally, to address the limitations of the Transformer architecture in handling long sequences, we propose the Hybrid-Transformer-Mamba (HTM), which combines the linear complexity and long-range modeling capabilities of Mamba with the Transformer's strength in modeling short sequences. We integrate these two model-agnostic modules into various existing methods and conduct experiments on seven datasets. The results demonstrate that incorporating these modules effectively overcomes the lookback window limitations. Notably, by combining them with the Patch strategy, we design the PIH (**P**atch-**I**BF-**H**TM), successfully extending the window length to 1024—a significantly larger window than previously achieved—and achieving state-of-the-art results, highlighting the potential of exploring even longer windows.

## 1 Introduction

Long-term time series forecasting (LTSF) (Lim & Zohren, 2020) holds significant importance across various domains such as traffic management, energy optimization, and financial analysis. Transformer-base methods (Vaswani et al., 2017), known for their attention mechanisms that facilitate the automatic learning of sequential dependencies, have emerged as promising tools for LTSF. Notable models like Informer (Zhou et al., 2021), Autoformer (Wu et al., 2021), and PatchTST (Nie et al., 2023) have demonstrated successful applications of Transformers in this domain. To enhance the forecasting capability of the model, extending the lookback window is a natural choice. A longer window enables the model to capture long-term trends more accurately, improving its ability to predict seasonal variations, cyclical patterns, and overall trends. For example, as shown in Fig. 1 (a), when using a longer window $L_2$, the model successfully captures the cyclical trend in the highlighted elliptical region, whereas using a shorter window $L_1$ results in failure. In theory, as the window length $L$ increases, the model's performance should gradually improve. However, current Transformer-based models encounter a Lookback Window Limitation (LWL) (Zeng et al., 2022). This limitation implies that after reaching the optimal performance at a certain window length $L$, further increasing the window does not yield better results. A natural question then arises: *How can we break through LWL and enable the model to perform better with longer windows?*

We analyse this issue from both an information-theoretic perspective and a model architecture perspective. **From the information perspective**, time series naturally possess redundancy, and longer windows tend to have higher redundancy (Prichard & Theiler, 1994a;b). As shown in Fig. 1 (b), after

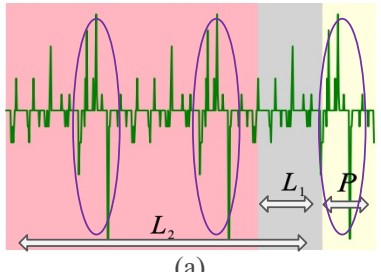 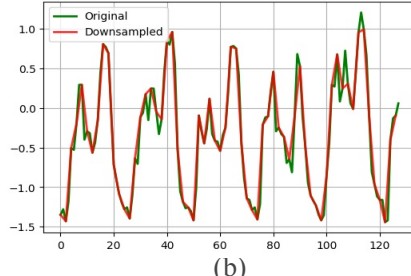

Figure 1: **(a)**: When predicting $P$ using a smaller lookback window $L_1$, the information regarding the elliptical part is not captured, resulting in inaccurate predictions. In contrast, longer window $L_2$ can capture the periodicity of the elliptical part. **(b)**: The redundancy in temporal information is evident from the fact that both the original sequence (green) and the downsampled sequence (red) maintain almost identical temporal characteristics.

downsampling the original sequence, the subsequences can still maintain almost identical temporal characteristics. Longer windows exacerbate this redundancy, as illustrated in Fig. 1 (a). Although $L_2$ provides more historical information, the several subsequences formed by elliptical segmentation are highly repetitive, resulting in $L_2$ having significantly higher redundancy than $L_1$. Therefore, although larger windows provide more information, the high level of redundancy can interfere with the model's learning. **From the model architecture perspective**, despite the Transformer's powerful sequence modeling capabilities, recent research (Zeng et al., 2022) has indicated that it tends to overfit temporal noises rather than extract temporal information when presented with longer sequences. Additionally, the quadratic complexity of the Transformer also hinders the exploration of longer windows.

The Patch strategy is one approach to overcome LWL by treating consecutive time steps as a single patch (Nie et al., 2023; Zhang & Yan, 2023). This reduces sequence redundancy and significantly decreases the effective sequence length for the Transformer. However, the Patch method is heuristic and lacks adaptability. It can only reduce redundancy at the local level, failing to address redundancy at the global level. Moreover, it does not mitigate the quadratic complexity inherent in Transformers. Moreover, it does not mitigate the quadratic complexity inherent in Transformers. As the number of patches increases, the computational demands increase dramatically.

In this paper, we propose two model-agnostic modules to address the issues of information redundancy and architectural limitations, respectively. **To alleviate information redundany**, we introduce the Information Bottleneck Filter (IBF) module based on information bottleneck (IB) theory. The IBF module aims to identify informative subsequences while minimizing redundancy and noise (Alemi et al., 2016), enabling the model to prioritize significant subsequences within the sequence. Directly optimizing the IB objective for sequences proves challenging owing to their discrete nature (Yu et al., 2021b;a), often resulting in training instability and degraded outcomes. Here, we propose the adoption of a probabilistic framework for sequence selection, alongside the introduction of a noise injection strategy. Initially, noise is injected into sequence elements with a certain probability, thereby disrupting the flow of information from the input sequence to the perturbed sequence. Subsequently, we incentivize the perturbed sequence to retain its informative properties in relation to the labels. The fundamental concept underlying this approach is that important subsequences should have a low probability of noise injection, whereas injecting larger noise into redundant sequences does not significantly impact predictions. By tailoring a noise prior for each input, the IB objective can yield a manageable variational upper bound. **To address the difficulties that Transformers face in handling long sequences**, we introduce Mamba (Gu & Dao, 2023), a recently proposed State Space Model (SSM) characterized by linear complexity. Mamba has garnered attention for its efficacy and efficiency in modeling extensive dependencies within sequential data (Ma et al., 2024; Liu et al., 2024b; Wang et al., 2024), rendering it particularly suitable for temporal data analysis. However, this does not imply a complete replacement of Transformers with Mamba. On one hand, while Mamba theoretically demonstrates linear complexity, Transformers incur lower computational overheads for shorter sequences owing to efficient hardware optimizations (see Appendix A.6). On the other hand, in short sequence modeling, we observe discernible performance differences between Transformers and Mamba across various datasets, potentially stemming from their distinct capabilities in encoding diverse sequence patterns. To harness the strengths of both ar-

chitectures simultaneously, we propose Hybrid-Transformer-Mamba (HTM). Specifically, rooted in the unique characteristics of time series data where temporal relationships persist even after down-sampling, we partition lengthy sequences into shorter subsequences. Then, we employ Mamba to capture long-term information from the input long sequence, while utilizing Transformer to capture short-term information from the short subsequences.

We integrated the two aforementioned model-agnostic modules into multiple Transformer-based models and conducted detailed experiments on seven datasets. The results demonstrate that these modules can effectively assist Transformer-based models in overcoming the LWL, enabling better performance with larger windows while reducing computational costs by 2 to 3 times. Notably, by incorporating these modules into the PatchTST model, we developed the PIH model (**P**atch-**IBF**-**H**TM), where the window length was extended to 1024—a significantly larger setting than in previous studies. The PIH model achieved state-of-the-art results, proving the effectiveness of using longer lookback windows. Our work can inspire future research to explore even longer window sizes. (Recent time series large models (Liu et al., 2024a; Jin et al., 2024) have adopted window sizes greater than $L = 1024$, which we will discuss in Appendix A.1 in relation to our approach.)

In summary, our primary contributions are as follows: First, while previous work has identified the existence of the LWL in Transformer-based methods, we focus on overcoming this limitation. Secondly, we introduce IBF and HTM, two model-agnostic modules designed from the perspectives of the information bottleneck and model architecture, respectively, to address the LWL. Thirdly, by integrating these modules into multiple existing models, we observe substantial performance improvements across seven datasets. Notably, the PIH model, which combines these modules with the Patch strategy, achieved state-of-the-art results, demonstrating the effectiveness and versatility of our proposed modules.

## 2 RELATED WORK

### 2.1 TRANSFORMER-BASED MODELS

Due to the attention mechanism's capability to capture long-range dependencies, Transformer-based models have found widespread application in language and vision tasks. Early attempts (Song et al., 2018; Ma et al., 2019; Li et al., 2019) at directly applying vanilla Transformers to time series data failed in long sequence forecasting tasks, as the self-attention operation scales quadratically with the input sequence length. Existing approaches primarily address this challenge through two avenues. Patch-based methods, exemplified by PatchTST (Nie et al., 2023) and CrossFormer (Zhang & Yan, 2023), conceptualize consecutive time steps as patches, reducing the number of input tokens and augmenting local semantics to mitigate redundancy. However, patch-based methods impose constraints on the input data format, and computational expenses persist even at the patch level when the window is large. Another approach focuses on sparse attention mechanisms. Models such as Informer (Zhou et al., 2021), Autoformer (Wu et al., 2021), Pyraformer (Liu et al., 2022b), and FEDformer (Zhou et al., 2022) adapt the self-attention mechanism to achieve complexities of $O(L)$ or $O(L \log(L))$. These models rely on specific designs and often sacrifice representational capacity, thereby compromising performance. Our work is independent of these approaches and can be effectively integrated into them.

### 2.2 MAMBA FOR TIME SERIES

Recently, several approaches have emerged to incorporate Mamba into time series modeling, each introducing unique innovations to enhance the capture of temporal dynamics. Bi-Mamba+ (Liang et al., 2024) introduces a novel Mamba+ block by incorporating a forget gate within Mamba. This modification enables the selective combination of new features with historical ones in a complementary manner, boosting the model's ability to balance past and present information. To further enhance feature interactions among time series elements, Bi-Mamba+ applies this approach in both forward and backward directions. S-Mamba (Wang et al., 2024) adopts a different approach by autonomously tokenizing time points of each variate using a linear layer. The method employs a bidirectional Mamba layer to extract inter-variate correlations and a Feed-Forward Network to learn temporal dependencies. Ultimately, S-Mamba generates forecasting results through a linear mapping layer, highlighting its structured yet flexible approach to capturing temporal patterns. TimeMa-

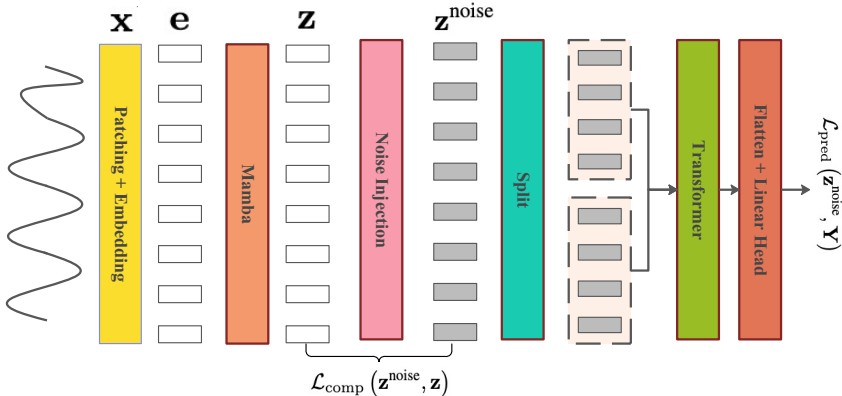

Figure 2: Overall of PIH architecture

chine (Ahamed & Cheng, 2024) takes a broader view of time series data by leveraging multi-scale contextual cues. Its architecture integrates a quadruple-Mamba design, allowing the model to manage both channel-mixing and channel-independence scenarios. By unifying global and local contexts at varying scales, TimeMachine effectively selects key information for prediction, thus offering robust handling of complex temporal structures. MambaTS (Cai et al., 2024) challenges the necessity of causal convolution within Mamba for long-term series forecasting (LTSF). It proposes the Temporal Mamba Block (TMB) as an alternative. To further prevent model overfitting, MambaTS incorporates a dropout mechanism that selectively applies to TMB's parameters, ensuring a more stable and generalizable model performance.

### 2.3 INFORMATION BOTTLENECK (IB)

The essence of the IB principle lies in distilling a compact yet predictive code from the input signal (Tishby et al., 2000). Pioneering work by (Alemi et al., 2016) introduced the concept of variational information bottleneck (VIB), thereby enriching deep learning methodologies. Presently, IB and VIB find extensive applications in deep learning, predominantly in representation learning and feature selection domains. In representation learning, the focus is on training deterministic or stochastic encoders to derive condensed yet semantically rich representations of input data. These representations serve as valuable inputs for a plethora of downstream tasks spanning computer vision (Luo et al., 2019; Peng et al., 2019), reinforcement learning (Goyal et al., 2019; Igl et al., 2019), natural language processing (Wang et al., 2020), and node representation learning (Wu et al., 2020). Meanwhile, in the realm of feature selection, IB is used to select a subset of input features such as pixels in images or dimensions in vectors, which are maximally predictive to the label of input data. Strategies such as injecting noise into intermediate representations of pre-trained networks and subsequently selecting regions with optimal information per dimension have been explored (Achille & Soatto, 2018; Schulz & et al., 2020). Additionally, techniques like learning drop rates for individual dimensions of vector-structured features have been proposed (Kim et al., 2021).

## 3 METHOD

Given a collection of multivariate time series samples with lookback window $L : (\mathbf{x}_1, \ldots, \mathbf{x}_L)$ where each $\mathbf{x}_t$ at time step $t$ is a vector of dimension $C$, we would like to forecast $T$ future values $(\mathbf{x}_{L+1}, \ldots, \mathbf{x}_{L+T})$. We integrate HTM and IBF into the PatchTST framework, resulting in PIH, as illustrated in Fig. 2. It is worth noting that our method is model-agnostic. In section 4, we also discuss its integration into other Transformer-based models.

### 3.1 PATCHING

Given our utilization of a channel-independent strategy, we opt for simplicity by converting multivariate time series into univariate ones. The input univariate time series $\mathbf{x}$ is initially segmented into patches, which may be either overlapping or non-overlapping. Employing patching strategies

enhances locality and captures comprehensive semantic information beyond the point level by aggregating time steps into subseries-level patches. Furthermore, to ensure uniform partitioning of the patch sequence into $K$ equally-sized blocks in subsequent modules (refer to section 3.3), we employ $\text{Padding}(\cdot)$ to extend the input sequence. Denoting the patch length as $P$ and the stride (the non-overlapping region between two consecutive patches) as $S$, the $\text{Patch}(\cdot)$ process yields a sequence of patches $\mathbf{h} \in \mathbb{R}^{N \times P}$, where $N$ denotes the number of patches, $N = \lceil \frac{(L-P)}{SK} \rceil * K$. Subsequently, we employ an embedding layer to map the dimension of each patch from $\mathbf{h} \in \mathbb{R}^{N \times P}$ to $\mathbf{e} \in \mathbb{R}^{N \times d}$.

$$\mathbf{e} = \text{Embedding}\left(\text{Patch}\left(\text{Padding}\left(\mathbf{x}\right)\right)\right) \tag{1}$$

## 3.2 Information Bottleneck Filter (IBF) Module for Redundancy Filtering

After obtaining the patch embedding sequence $\mathbf{e} = \{e_1, e_2, \ldots, e_N\}$, our approach involves the application of Mamba, followed by a subsequent Dropout layer to capture long-term dependency:

$$\mathbf{z} = \text{Dropout}(\text{Mamba}(\mathbf{e})) \tag{2}$$

In scenarios where the patch sequence length $N$ is considerable, there exists a possibility of significant redundancy. To address this issue, we leverage the information bottleneck theory to filter out redundant information of $\mathbf{z}$.

**Information Bottleneck (IB).** In machine learning, determining which aspects of input data to retain and which to discard is crucial. The Information Bottleneck (IB) principle (Alemi et al., 2016) offers a systematic approach to this by compressing the source random variable to preserve information relevant for predicting the target random variable, while discarding irrelevant information. Given random variables $X$ and $Y$, IB aims to compress $X$ into a bottleneck random variable $B$, while retaining information pertinent to predicting $Y$:

$$\min_{B} -I(Y; B) + \beta I(X; B) \tag{3}$$

Here, $\beta$ serves as a Lagrangian multiplier to balance the two mutual information terms.

**Rationale for filtering information from z instead of directly from e:** Mamba can be conceptualized as a variant of recurrent neural networks (Hochreiter & Schmidhuber, 1997; Schuster & Paliwal, 1997). Therefore, the representation $\mathbf{z}_t$ of the $t$-th patch in Mamba accumulates information not only from the current patch $\mathbf{e}_t$, but also from historical data $[\mathbf{e}_1, \ldots, \mathbf{e}_{t-1}]$. In contrast, $\mathbf{e}_t$ solely contains information from the current patch. Considering the temporal nature of time series data, the importance of the $t$-th patch is influenced not only by its own state but also by preceding patches. Therefore, we apply IBF after Mamba layers.

The IBF module seeks to retrieve the most relevant subsequence $\mathbf{x}^{\text{sub}}$ for a target prediction $\mathbf{Y}$ from the input sequence $\mathbf{x}$. We adopt the sufficient encoder assumption (Tian et al., 2020), implying that the information of the input subsequence $\mathbf{x}^{\text{sub}}$ is preserved in the encoding process, resulting in $I(\mathbf{x}^{\text{sub}}, \mathbf{Y}) \approx I(\mathbf{z}^{\text{sub}}, \mathbf{Y})$ and $I(\mathbf{x}^{\text{sub}}, \mathbf{x}) \approx I(\mathbf{z}^{\text{sub}}, \mathbf{z})$, where $\mathbf{z}^{\text{sub}}$ is a subsequence of $\mathbf{z}$. The Eq. 3 are transformed into:

$$\min_{\mathbf{z}^{\text{sub}}} -I(\mathbf{z}^{\text{sub}}, \mathbf{Y}) + \beta I(\mathbf{z}^{\text{sub}}, \mathbf{z}) \tag{4}$$

The first term encourages $\mathbf{z}^{\text{sub}}$ to be informative to the label $\mathbf{Y}$ and the second term minimizes the mutual information of $\mathbf{z}$ and $\mathbf{z}^{\text{sub}}$, so that $\mathbf{z}^{\text{sub}}$ only receives limited information from $\mathbf{z}$. The discrete nature of sequences renders direct optimization of IB objective impractical, as there are $2^N$ potential subsequences $\mathbf{z}^{\text{sub}}$ for a patch sequence of length $N$. To address this challenge, we relax patch weights from binary to continuous variables within the range $(0, 1)$. Considering $\mathbf{z}_i$ as the representation of the $i$-th patch, encapsulating information up to and including the $i$-th patch, we utilize MLP to assess the importance $\mathbf{c}_i$ of patch $\mathbf{z}_i$:

$$\mathbf{c}_i = \text{sigmoid}\left(\text{MLP}\left(\mathbf{z}_i\right)\right) \tag{5}$$

Consequently, the selection of patch $\mathbf{z}_i$ can be obtained by sampling from $\lambda_i \sim \text{Bern}(\mathbf{c}_i)$, where $\text{Bern}(\mathbf{c}_i)$ represents a Bernoulli distribution parameterized by $\mathbf{c}_i$. To ensure the differentiability of the sampling process, we utilize the gumbel sigmoid (Maddison et al., 2017; Jang et al., 2017) function for the discrete random variable $\lambda_i$, defined as:

$$\lambda_i = \text{Sigmoid}\left(\frac{1}{\tau}\log\left[\frac{\mathbf{c}_i}{1 - \mathbf{c}_i}\right] + \log\left[\frac{u}{1-u}\right]\right) \tag{6}$$

where $u \sim \text{Uniform}(0, 1)$, and $\tau$ is the temperature hyperparameter. Subsequently, subsequence $\mathbf{z}^{\text{sub}}$ can be obtained by $\mathbf{z}^{\text{sub}} = \lambda\mathbf{z}$. Although we can employ shannon mutual information (Duncan, 1970) to quantify the compressed and informative nature of the distribution of subsequences $\mathbf{z}^{\text{sub}}$, the optimization process is inefficient and unstable due to mutual information estimation (Yu et al., 2021b). To address this challenge, we employ an optimization strategy known as noise injection (Yu et al., 2021a), which consists of two stages: sequence perturbation and sequence selection. The core concept is to allow the model to introduce noise into less informative subsequences while minimizing noise injection into more informative ones. Initially, noise injection disrupts the flow of information from the input sequence $\mathbf{z}$ to the perturbed sequence $\mathbf{z}^{\text{noise}}$. Subsequently, we encourage the perturbed sequence $\mathbf{z}^{\text{noise}}$ to maintain its informative properties relative to the label $\mathbf{Y}$. Finally, $\mathbf{z}^{\text{sub}}$ is derived by removing the noise from $\mathbf{z}^{\text{noise}}$. Eq. 4 can be reformulated as:

$$\min_{\mathbf{z}^{\text{noise}}} -I(\mathbf{z}^{\text{noise}}, Y) + \beta I(\mathbf{z}^{\text{noise}}, \mathbf{z}) \tag{7}$$

where $\mathbf{z}^{\text{noise}} = \lambda\mathbf{z} + (1 - \lambda)\epsilon$, and $\epsilon$ follows a random Gaussian distribution. To preserve the semantic of $\mathbf{z}^{\text{noise}}$, we set $\epsilon \sim \mathcal{N}(\mu_{\mathbf{z}}, \sigma_{\mathbf{z}}^2)$, where $\mu_{\mathbf{z}}$ and $\sigma_{\mathbf{z}}^2$ denote the mean and variance of $\mathbf{z}$. We first examine the first term $-I(\mathbf{z}^{\text{noise}}, \mathbf{Y})$ in Eq. 7 which encourages $\mathbf{z}^{\text{noise}}$ is informative of label $\mathbf{Y}$:

$$-I(\mathbf{z}^{\text{noise}}, \mathbf{Y}) \leq \mathbb{E}_{\mathbf{Y}, \mathbf{z}^{\text{noise}}} - \log p_\theta(\mathbf{Y} \mid \mathbf{z}^{\text{noise}}) := \mathcal{L}_{\text{pred}}(\mathbf{z}^{\text{noise}}, \mathbf{Y}) \tag{8}$$

Here, $p_\theta(\mathbf{Y} \mid \mathbf{z}^{\text{noise}})$ represents the variational approximation to the true posterior distribution $p(\mathbf{Y} \mid \mathbf{z}^{\text{noise}})$ (A detailed proof can be found in Appendix A.4). We model $p_\theta(\mathbf{Y} \mid \mathbf{z}^{\text{noise}})$ as a predictor parametrized by $\theta$, which outputs the model prediction $\mathbf{Y}$ based on the input $\mathbf{z}^{\text{noise}}$. Thus, we can minimize the upper bound of $-I(\mathbf{z}^{\text{noise}}, \mathbf{Y})$ by minimizing the model prediction loss $\mathcal{L}_{\text{pred}}(\mathbf{z}^{\text{noise}}, \mathbf{Y})$. We choose to utilize the Mean Squared Error (MSE) loss as $\mathcal{L}_{\text{pred}}(\mathbf{z}^{\text{noise}}, \mathbf{Y})$.

For the second term $I(\mathbf{z}^{\text{noise}}, \mathbf{z})$ in Eq. 7, we can derive its variational upper bound:

$$-I(\mathbf{z}^{\text{noise}}, \mathbf{z}) \leq \mathbb{E}_{\mathbf{z}} \left( -\frac{1}{2}\log A + \frac{1}{2N}A + \frac{1}{2N}B^2 \right) := \mathcal{L}_{\text{comp}}(\mathbf{z}^{\text{noise}}, \mathbf{z}) \tag{9}$$

where $A = \sum_{j=1}^{N}(1 - \lambda_j)^2$ and $B = \frac{\sum_{j=1}^{N} \lambda_j(\mathbf{z}_j - \mu_{\mathbf{z}})}{\sigma_{\mathbf{z}}}$. A detail proof is given in Appendix A.4.

Finally, we can efficiently estimate Eq. 8 and Eq. 9 with the batched data in the training set. The overall loss is:

$$\mathcal{L} = \mathcal{L}_{\text{pred}}(\mathbf{z}^{\text{noise}}, \mathbf{Y}) + \beta\mathcal{L}_{\text{comp}}(\mathbf{z}^{\text{noise}}, \mathbf{z}) \tag{10}$$

### 3.3 HYBRID-TRANSFORMER-MAMBA(HTM)

Modeling the input long sequence with Mamba and then using Transformer to model the partitioned short sequences is a promising paradigm (Mehta et al., 2023; Pilault et al., 2023; Lieber et al., 2024), as it can leverage the strengths of both architectures simultaneously. We have designed two split methods capable of retaining semantic information: *interval split* and *block split*, denoted as:

$$b_i = \{\mathbf{z}_j^{\text{noise}} \in \mathbf{z}^{\text{noise}} : i \equiv j \pmod{K}\} \tag{11}$$

$$b_i = \mathbf{z}_{(i-1)*N/K:i*N/K}^{\text{noise}} \tag{12}$$

where $b_i$ represents the $i$-th sequence block, and $K$ is the number of blocks. The premise for splitting sequences into subsequences is that the latter can still retain the semantic meaning of the original long sequences. Fortunately, time series data often adhere to this principle. The *interval split* is inspired by SCINet (Liu et al., 2022a), which highlights a unique property of time series: temporal relations (e.g., trend and seasonal components) are largely preserved after downsampling into two subsequences. SCINet downsamples the original sequence into two subsequences by separating the even and odd elements, our *interval split* extends this approach to partitioning patch sequence into $K$ blocks, distributing contiguous $K$ patches into $K$ distinct blocks. This partitioning method preserves the global characteristics of the sequence. Additionally, we propose the *block split*, where a continuous segment of patch subsequence forms a block. This partitioning method is based on the periodicity of time series, where one period (or multiples of a period) is considered as a block, thus preserving the local information of the sequence.

The patch operation and partitioning reduce the length of the input sequence for Transformer from $L$ to $L/PK$, significantly reducing the computational overhead. Combined with Mamba processing the entire sequence, the overall time complexity of the Hybrid Transformer Model (HTM) becomes $O(L/P) + O((L/PK)^2)$. Although the latter term still exhibits quadratic complexity, appropriate choices of $P$ and $K$ can maintain $L/P$ within an acceptable constant range.

## 4 EXPERIMENT

Our experiments are divided into three parts. In the first part, we set the lookback window length to $L = 1024$, which, to our knowledge, is longer than any previously used method. The PIH model achieved state-of-the-art results, encouraging future research to explore even longer windows. Additionally, the IBF module enhances the model's interpretability, while the HTM module significantly reduces computational costs. In the second part, we investigate the integration of the IBF and HTM modules into other Transformer-based models, such as Transformer, Informer, and Autoformer. The results demonstrate that, after incorporating these modules, the models effectively overcome LWL and achieve better performance with longer windows, highlighting the general applicability of these modules. Future research could adopt these model-agnostic modules to improve performance with extended windows. Finally, in the third part, we conducted ablation experiments on the model components.

### 4.1 COMPARISON OF PIH WITH OTHER MODELS

**Experimental Settings and Baselines.** We evaluate PIH on seven popular datasets (See Appendix A.2), including Weather, Traffic, Electricity, and four ETT datasets (Etth1, Etth2, Ettm1, Ettm2). PIH integrates the IBF and HTM modules into the PatchTST model, making PatchTST the primary baseline. To assess how effectively our model utilizes longer lookback windows, we set $L = 1024$ for both PIH and PatchTST, which is significantly longer than in previous studies. The other experimental settings can be found in Appendix A.5.

We additionally selected Mamba-based, Transformer-based, and Linear-based models as baselines. S-Mamba (Wang et al., 2024) utilizes a bidirectional Mamba layer to extract inter-variate correlations, while a Feed-Forward Network is employed to learn temporal dependencies. For Transformer-based models, in addition to PatchTST, we selected three other models: FEDformer (Zhou et al., 2022), Autoformer (Wu et al., 2021), and Informer (Zhou et al., 2021). Since these baselines were originally designed with relatively shorter windows (e.g., 96), we reran them with seven different lookback windows $L = \{24, 48, 96, 192, 336, 720, 1024\}$ and selected the best results to establish robust baselines. Furthermore, we include two Linear-based models, DLinear and NLinear (Zeng et al., 2022). Given that these two models were proposed to address the limitations of Transformer-based models in handling long lookback windows, we also set $L = 1024$ for them. All models follow the same experimental setup, with prediction lengths $T \in \{96, 192, 336, 720\}$. We use MSE and MAE as evaluation metrics.

**Results and Analysis.** The results of multivariate long-term forecasting are summarized in Tab. 1. For models like S-Mamba, Transformer, Autoformer, and Informer, PIH significantly outperforms them. Even for models specifically designed to handle long sequences, such as PatchTST, DLinear, and NLinear, PIH still surpasses them, demonstrating its effectiveness in processing longer sequences. It is worth noting that we did not intentionally choose an unusual setting like $L = 1024$ to lower the performance of these three models. In Appendix A.3, we also provide their performance under shorter windows (e.g., 336 and 512), where PIH continues to outperform them. Overall, PIH with a much longer window setting achieves better results than other models with shorter windows. Our experiments highlight the potential for further increasing the window size.

**The Potential of Longer Windows.** Tab. 1 shows that under long lookback window settings with $L = 1024$, PIH significantly outperforms other methods. We further explore whether expanding the window size is meaningful. As shown in Fig. 3 (a), we set the lookback window to $L = \{96, 336, 512, 1024\}$ and used the average MSE over 7 datasets with forecasting horizons of $T \in \{96, 192, 336, 720\}$ as the evaluation metric. The results indicate that the performance of PatchTST improves steadily as the window increases from 96 to 512, but declines when extended

Table 1: Multivariate long-term forecasting results with different prediction lengths $T \in \{96, 192, 336, 720\}$. We provide the mean value for each column in the final row.

| Models | PIH | | PatchTST | | S-Mamba | | FEDformer | | Autoformer | | Informer | | DLinear | | NLinear | |
|---|---|---|---|---|---|---|---|---|---|---|---|---|---|---|---|---|
| Metric | MSE | MAE | MSE | MAE | MSE | MAE | MSE | MAE | MSE | MAE | MSE | MAE | MSE | MAE | MSE | MAE |
| ETTh1 96 | **0.360** | **0.394** | 0.371 | 0.405 | 0.386 | 0.406 | 0.376 | 0.415 | 0.435 | 0.446 | 0.941 | 0.769 | 0.511 | 0.520 | 0.379 | 0.404 |
| ETTh1 192 | **0.396** | **0.418** | 0.408 | 0.429 | 0.448 | 0.444 | 0.423 | 0.446 | 0.456 | 0.457 | 1.007 | 0.786 | 0.414 | 0.428 | 0.414 | 0.426 |
| ETTh1 336 | **0.409** | **0.432** | 0.431 | 0.449 | 0.494 | 0.468 | 0.444 | 0.462 | 0.486 | 0.487 | 1.038 | 0.784 | 0.453 | 0.458 | 0.442 | 0.445 |
| ETTh1 720 | **0.435** | **0.466** | 0.482 | 0.483 | 0.493 | 0.488 | 0.469 | 0.492 | 0.515 | 0.517 | 1.144 | 0.857 | 0.511 | 0.520 | 0.470 | 0.477 |
| ETTh2 96 | **0.263** | **0.328** | 0.277 | 0.340 | 0.298 | 0.349 | 0.332 | 0.374 | 0.332 | 0.368 | 1.549 | 0.952 | 0.294 | 0.361 | 0.296 | 0.351 |
| ETTh2 192 | **0.324** | **0.370** | 0.343 | 0.385 | 0.379 | 0.398 | 0.407 | 0.446 | 0.426 | 0.434 | 3.792 | 1.542 | 0.430 | 0.448 | 0.337 | 0.382 |
| ETTh2 336 | **0.314** | **0.376** | 0.338 | 0.394 | 0.417 | 0.432 | 0.400 | 0.447 | 0.477 | 0.479 | 4.215 | 1.642 | 0.492 | 0.484 | 0.359 | 0.407 |
| ETTh2 720 | **0.378** | **0.425** | 0.403 | 0.442 | 0.431 | 0.449 | 0.412 | 0.469 | 0.453 | 0.490 | 3.656 | 1.619 | 0.905 | 0.683 | 0.417 | 0.456 |
| ETTm1 96 | **0.291** | **0.349** | 0.294 | 0.349 | 0.331 | 0.368 | 0.326 | 0.390 | 0.510 | 0.492 | 0.626 | 0.560 | 0.314 | 0.358 | 0.317 | 0.359 |
| ETTm1 192 | 0.337 | **0.374** | **0.334** | **0.374** | 0.371 | 0.387 | 0.365 | 0.415 | 0.514 | 0.495 | 0.725 | 0.619 | 0.356 | 0.391 | 0.352 | 0.381 |
| ETTm1 336 | **0.360** | **0.386** | 0.363 | 0.392 | 0.417 | 0.418 | 0.392 | 0.425 | 0.510 | 0.492 | 1.005 | 0.741 | 0.365 | 0.388 | 0.374 | 0.393 |
| ETTm1 720 | **0.405** | **0.411** | 0.407 | 0.416 | 0.471 | 0.448 | 0.446 | 0.458 | 0.527 | 0.493 | 1.133 | 0.845 | 0.410 | 0.417 | 0.409 | 0.413 |
| ETTm2 96 | **0.161** | **0.253** | 0.164 | 0.259 | 0.179 | 0.263 | 0.180 | 0.271 | 0.205 | 0.293 | 0.355 | 0.462 | 0.164 | 0.260 | 0.163 | 0.257 |
| ETTm2 192 | **0.213** | **0.289** | 0.216 | 0.295 | 0.253 | 0.310 | 0.252 | 0.318 | 0.278 | 0.336 | 0.595 | 0.586 | 0.238 | 0.317 | 0.216 | 0.294 |
| ETTm2 336 | **0.265** | **0.326** | 0.268 | 0.331 | 0.312 | 0.348 | 0.324 | 0.364 | 0.343 | 0.379 | 1.270 | 0.871 | 0.265 | 0.326 | 0.265 | 0.326 |
| ETTm2 720 | **0.342** | **0.375** | 0.350 | 0.383 | 0.412 | 0.408 | 0.410 | 0.420 | 0.414 | 0.419 | 3.001 | 1.267 | 0.338 | 0.375 | 0.338 | 0.375 |
| Weather 96 | **0.147** | 0.198 | **0.147** | **0.197** | 0.166 | 0.210 | 0.238 | 0.314 | 0.249 | 0.329 | 0.354 | 0.405 | 0.167 | 0.225 | 0.170 | 0.226 |
| Weather 192 | 0.191 | **0.239** | **0.190** | 0.241 | 0.215 | 0.253 | 0.275 | 0.329 | 0.325 | 0.370 | 0.419 | 0.434 | 0.211 | 0.267 | 0.215 | 0.265 |
| Weather 336 | **0.241** | **0.280** | 0.243 | 0.283 | 0.276 | 0.298 | 0.339 | 0.377 | 0.351 | 0.391 | 0.583 | 0.543 | 0.255 | 0.304 | 0.259 | 0.298 |
| Weather 720 | 0.309 | 0.329 | **0.306** | **0.328** | 0.353 | 0.349 | 0.389 | 0.409 | 0.415 | 0.426 | 0.916 | 0.705 | 0.313 | 0.351 | 0.321 | 0.342 |
| Traffic 96 | **0.357** | **0.248** | 0.394 | 0.289 | 0.381 | 0.261 | 0.576 | 0.359 | 0.597 | 0.371 | 0.733 | 0.410 | 0.385 | 0.275 | 0.383 | 0.270 |
| Traffic 192 | **0.371** | **0.255** | 0.407 | 0.295 | 0.397 | 0.267 | 0.610 | 0.380 | 0.607 | 0.382 | 0.777 | 0.435 | 0.397 | 0.279 | 0.397 | 0.274 |
| Traffic 336 | **0.392** | **0.261** | 0.422 | 0.302 | 0.423 | 0.276 | 0.608 | 0.375 | 0.623 | 0.387 | 0.776 | 0.434 | 0.412 | 0.288 | 0.410 | 0.281 |
| Traffic 720 | **0.430** | **0.282** | 0.46 | 0.319 | 0.458 | 0.300 | 0.621 | 0.375 | 0.639 | 0.395 | 0.827 | 0.466 | 0.450 | 0.309 | 0.449 | 0.303 |
| Electricity 96 | **0.127** | **0.220** | 0.133 | 0.226 | 0.142 | 0.238 | 0.186 | 0.302 | 0.196 | 0.313 | 0.304 | 0.393 | 0.132 | 0.229 | 0.133 | 0.229 |
| Electricity 192 | **0.145** | **0.240** | 0.151 | 0.249 | 0.169 | 0.267 | 0.197 | 0.311 | 0.211 | 0.324 | 0.327 | 0.417 | 0.146 | 0.243 | 0.148 | 0.242 |
| Electricity 336 | **0.160** | **0.256** | 0.167 | 0.263 | 0.178 | 0.275 | 0.213 | 0.328 | 0.214 | 0.327 | 0.333 | 0.422 | 0.161 | 0.260 | 0.164 | 0.259 |
| Electricity 720 | **0.192** | **0.287** | 0.206 | 0.299 | 0.207 | 0.303 | 0.233 | 0.344 | 0.236 | 0.342 | 0.351 | 0.427 | 0.195 | 0.292 | 0.203 | 0.292 |
| Mean | **0.297** | **0.326** | 0.310 | 0.336 | 0.338 | 0.346 | 0.372 | 0.386 | 0.412 | 0.408 | 1.17 | 0.728 | 0.341 | 0.355 | 0.314 | 0.336 |

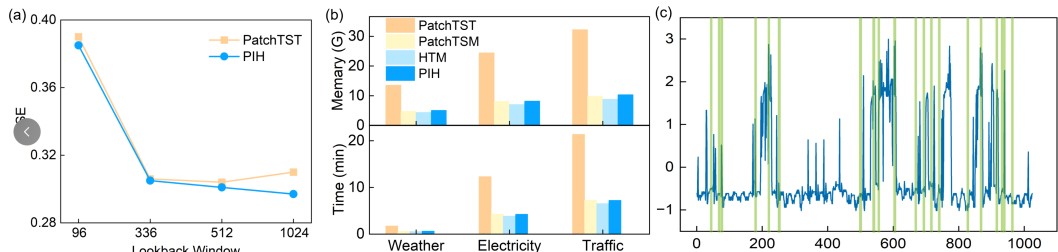

Figure 3: **(a)**: The performance comparison between PIH and PatchTST at $L \in \{96, 336, 512, 1024\}$. **(b)**: Comparison of GPU memory (GB) and training time (minutes/epoch) for PatchTST, PatchTST, HTM, and PIH. **(c)**: Visualization of a sample sequence in the *Electricity*, highlighting the most important 20 patches identified by the IBF module with green shading.

to 1024. In contrast, PIH exhibits a consistent performance improvement as the window size increases from 96 to 1024. This suggests that the HTM and IBF modules help PatchTST overcome the $L = 512$ window limitation, achieving better performance with longer windows. Another noteworthy observation is that, except for $L = 96$, PIH consistently outperforms PatchTST for the same $L$. We hypothesize that with $L = 96$, sequence redundancy is low, and the Patch strategy alone is sufficient to manage it effectively, rendering IBF and HTM unnecessary. Consequently, PIH lags behind PatchTST at this window size. However, as the window length increases and sequence redundancy grows, the IBF and HTM modules become more effective, allowing PIH to surpass PatchTST.

**Computational Overhead.** In addition to performance comparisons, we evaluated computation time and memory usage, as shown in Fig. 3 (b). When using only the HTM module without the IBF (referred to as HTM), it demonstrates significant improvements in both computational time and memory usage compared to the pure Transformer architecture (referred to as PatchTST), surpassing it by a notable margin (2 to 3 times). Additionally, HTM outperforms the pure Mamba architecture (referred to as PatchTSM), which can be attributed to the Transformer's lower computational cost when handling shorter sequences compared to Mamba. Moreover, when both HTM and IBF are integrated (i.e., PIH), the additional overhead introduced is negligible, as the IBF module only consists of a simple MLP.

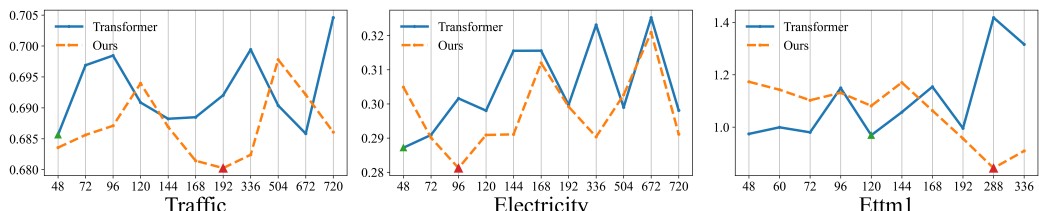

Figure 5: The performance changes across the Traffic, Electricity, and ETTm1 datasets upon integrating HTM and IBF into Transformers. The triangular markers indicate the window limitations.

**Interpretability of IBF.** Another advantage of incorporating the IBF module is its ability to enhance interpretability by identifying crucial subsequences for the final prediction. As shown in Fig. 3 (c), we provide a visualization of a sample from the *Electricity* dataset. The top 20 most important patches are marked in green, indicating that the model focuses more on sequences at peak positions.

### 4.2 INTEGRATION INTO OTHER MODELS.

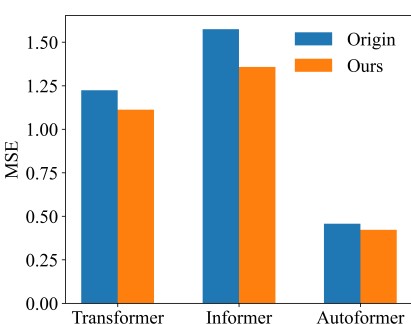

Figure 4: Performance comparison after integrating HTM and IBF into Transformer, Informer, and Autoformer.

We integrate the HTM and IBF modules into three different Transformer-based architectures to validate their generality (where "Origin" represents the original model and "Ours" denotes the integration of the HTM and IBF modules). We set various lookback windows $L = \{24, 48, 96, 192, 336, 720, 1024\}$ and a prediction length of $T = 720$, selecting the best results. We utilize the average MSE across seven datasets as the evaluation metric, with the results illustrated in Fig. 4. Informer, Autoformer, and Transformer all demonstrate significant performance improvements after incorporating the HTM and IBF modules. Additionally, we present the performance curves (MSE) for the ETTm1, Electricity, and Traffic datasets with a prediction length of $T = 720$ in Fig. 5. For the original Transformer models, the lookback window limitations for these three datasets are 48, 48, and 120, respectively, while our models increase these limitations to 192, 96, and 228, achieving better performance. Furthermore, we observe that with smaller windows, issues such as information redundancy and the inherent weaknesses of Transformers are less pronounced, leading to similar or even worse performance from our models. However, as the window size increases, our models significantly outperform the original Transformers.

### 4.3 ABLATION STUDY

**Component Ablation.** We introduce HTM module and IBF module. To assess their effectiveness, we utilize PatchTST as a baseline, upon which we separately introduce IBF, HTM and both simultaneously to obtain three variants: +IB, +HTM, and PIH. Additionally, we introduce a variant of HTM, HMM, which solely employs Mamba to handle both the original long sequences and the divided short sequences. We refrain from designing a variant that processes the original long sequences with Transformer and the divided short sequences with Mamba, as it contradicts our goal of reducing computational complexity. All experiments maintain consistent settings, with a lookback window set to 1024 and prediction lengths set to 96, 192, 336, and 720. The average MSE across seven datasets is used as the evaluation metric. As illustrated in Fig. 6, the following observations are made: (1) Both IBF and HTM modules enhance the model's performance, and combining these two modules yields superior results. (2) Compared to HMM, HTM exhibits slightly better performance, which can be attributed to the different mechanisms between Transformer and Mamba, making each more suited to handling different types of sequences. By combining the strengths of both, the hybrid approach achieves superior results. As discussed earlier, the Transformer has lower computational

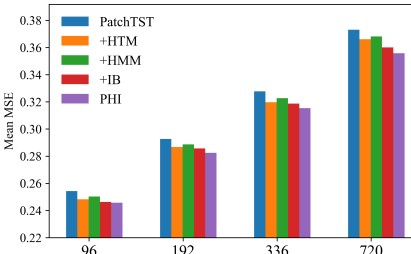 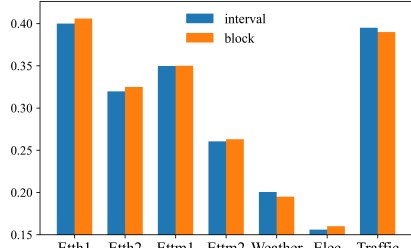

Figure 6: Left: Ablation experiments of different modules at prediction $T = \{96, 192, 336, 720\}$, using average MSE across 7 datasets as the evaluation metric. Right: Comparison of *interval split* and *block split* methods across different datasets, using average MSE across 7 datasets at prediction lengths $T = \{96, 192, 336, 720\}$ as the evaluation metric.

costs for shorter sequences, while Mamba is more efficient for longer sequences. Therefore, from both performance and computational overhead perspectives, using a combination of both architectures is a better choice than relying solely on one. (3) At longer prediction lengths, such as $T = 720$, our model demonstrates greater improvements compared to $T = 96$, indicating that larger windows $L$ provide more significant benefits for longer-term predictions (longer $T$).

**Interval Split vs. Block Split.** We compared the performance of *interval split* and *block split* across various datasets, as illustrated in Fig. 6. Overall, the effectiveness of both partitioning methods is roughly comparable, demonstrating their capability to preserve sequential characteristics. However, slight variations in performance are observed across different datasets. We speculate that this discrepancy arises from the distinct abilities of each partitioning method to retain specific sequential patterns. Intuitively, *interval split* emphasizes global variations, while *block split* focuses on variations within periods. Determining the most suitable partitioning strategy remains a subject for future investigation.

## 5    CONCLUSION AND FUTURE WORK

**Conclusion.**    In this paper, we focus on addressing the LWL by analyzing it from both model architecture and information-theoretic perspectives, proposing the HTM and IBF modules. We combine these with the patch strategy to design the PIH model, which can handle longer windows than previous works and achieves state-of-the-art results, demonstrating the potential of exploring longer windows. Additionally, we integrate these two modules into other Transformer-based models, enabling them to overcome window limitations and achieve improved performance with longer windows.

**Limitations and Future Work.**    First, our experiments demonstrate that extending the window length to $L = 1024$ still yields performance improvements, suggesting that further exploration of longer windows is a promising direction. Secondly, we found that longer lookback windows are not always beneficial for all datasets. Therefore, identifying which types of data are suitable for very long windows is another important area for future research. Thirdly, the *interval split* and *block split* methods proposed in this paper are heuristic. Designing an adaptive, end-to-end segmentation method tailored to each training dataset may lead to better results. Lastly, while recent large time-series models have adopted much longer windows, we claim that our approach is orthogonal to theirs. It is worth exploring whether our method can be integrated into these large models to further extend their window sizes.

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

## A Appendix

### A.1 Relationship with Large Time-Series Models

Although some recent large time-series models are capable of handling longer windows, they rely on significantly more parameters and much larger training datasets compared to our experiments. Additionally, when tested on the same datasets we used, these models still employ smaller window sizes. Our work does not conflict with these advancements in large time-series models. This is because the HTM and IBF modules we propose are model-agnostic and can be integrated into large time-series models, a direction worth exploring in future research.

### A.2 Dataset

We use 7 popular multivariate datasets provided in (Wu et al., 2021) for forecasting and representation learning. *Weather* dataset collects 21 meteorological indicators in Germany, such as humidity and air temperature. *Traffic* dataset records the road occupancy rates from different sensors on San Francisco freeways. *Electricity* is a dataset that describes 321 customers' hourly electricity consumption. *ETT*(Electricity Transformer Temperature) datasets are collected from two different electric transformers labeled with 1 and 2, and each of them contains 2 different resolutions (15 minutes and 1 hour) denoted with m and h. Thus, in total we have 4 ETT datasets: ETTm1, ETTm2, ETTh1, and ETTh2.

Table 2: Statistics of popular datasets for benchmark.

| Datasets | Weather | Traffic | Electricity | ETTh1 | ETTh2 | ETTm1 | ETTm2 |
|---|---|---|---|---|---|---|---|
| Features | 21 | 862 | 321 | 7 | 7 | 7 | 7 |
| Timesteps | 52696 | 17544 | 26304 | 17420 | 17420 | 69680 | 69680 |

### A.3 Performance of PatchTST, DLinear, and NLinear under Different Window Lengths

Here, we conducted experiments with DLinear and NLinear, two linear-based models, under two settings: $L = 336$ and $L = 1024$, with results shown in Table 3. For PatchTST, we do not present the results here because the original paper provides detailed results for PatchTST at window lengths of 336 and 512, while this paper includes results for a window length of 1024, making it unnecessary to repeat the information.We can draw the following conclusions:

- Linear-based models indeed perform well against noise, with NLinear($L = 1024$) generally outperforming NLinear($L = 336$). This is consistent with the results of PIH, indicating that larger windows are beneficial.

- NLinear($L = 1024$) generally outperforms NLinear($L = 336$), whereas DLinear($L = 1024$) consistently underperforms compared to DLinear($L = 336$). Thus, directly increasing the window size in linear-based methods is not always effective.

- PIH($L = 1024$) outperforms NLinear($L = 1024$), which can be attributed to the superior representational capabilities of the Transformer and Mamba modules compared to linear

modules. Therefore, it is essential to continue exploring the potential of Transformer-based models with longer windows rather than relying solely on linear-based models.

Table 3: Comparison between DLinear, NLinear, and PIH with lookback windows LL of 336 and 1024.

| | | Weather | | Traffic | | Electricity | | Etth1 | | Etth2 | | Ettm1 | | Ettm2 | | Avg. | | Total Avg. | |
|---|---|---|---|---|---|---|---|---|---|---|---|---|---|---|---|---|---|---|---|
| | | MSE | MAE | MSE | MAE | MSE | MAE | MSE | MAE | MSE | MAE | MSE | MAE | MSE | MAE | MSE | MAE | MSE | MAE |
| DLinear(336) | 96 | 0.176 | 0.237 | 0.410 | 0.282 | 0.140 | 0.237 | 0.375 | 0.399 | 0.289 | 0.353 | 0.299 | 0.343 | 0.167 | 0.260 | 0.265 | 0.302 | 0.332 | 0.351 |
| | 192 | 0.220 | 0.282 | 0.423 | 0.287 | 0.153 | 0.249 | 0.405 | 0.416 | 0.383 | 0.418 | 0.335 | 0.365 | 0.224 | 0.303 | 0.306 | 0.331 | | |
| | 336 | 0.265 | 0.319 | 0.436 | 0.296 | 0.169 | 0.267 | 0.439 | 0.443 | 0.448 | 0.465 | 0.369 | 0.386 | 0.281 | 0.342 | 0.344 | 0.360 | | |
| | 720 | 0.323 | 0.362 | 0.466 | 0.315 | 0.203 | 0.301 | 0.472 | 0.490 | 0.605 | 0.551 | 0.425 | 0.421 | 0.397 | 0.421 | 0.413 | 0.409 | | |
| DLinear(1024) | 96 | 0.167 | 0.225 | 0.385 | 0.275 | 0.132 | 0.229 | 0.378 | 0.403 | 0.294 | 0.361 | 0.314 | 0.358 | 0.164 | 0.260 | 0.262 | 0.301 | 0.341 | 0.355 |
| | 192 | 0.211 | 0.267 | 0.397 | 0.279 | 0.146 | 0.243 | 0.414 | 0.428 | 0.430 | 0.448 | 0.356 | 0.391 | 0.238 | 0.317 | 0.313 | 0.339 | | |
| | 336 | 0.255 | 0.304 | 0.412 | 0.288 | 0.161 | 0.260 | 0.453 | 0.458 | 0.492 | 0.484 | 0.365 | 0.388 | 0.265 | 0.326 | 0.343 | 0.358 | | |
| | 720 | 0.313 | 0.351 | 0.450 | 0.309 | 0.195 | 0.292 | 0.511 | 0.520 | 0.905 | 0.683 | 0.410 | 0.417 | 0.338 | 0.375 | 0.446 | 0.421 | | |
| NLinear(336) | 96 | 0.182 | 0.232 | 0.410 | 0.279 | 0.141 | 0.237 | 0.374 | 0.394 | 0.277 | 0.338 | 0.306 | 0.348 | 0.167 | 0.255 | 0.265 | 0.298 | 0.337 | 0.333 |
| | 192 | 0.225 | 0.269 | 0.410 | 0.279 | 0.154 | 0.248 | 0.408 | 0.415 | 0.344 | 0.381 | 0.349 | 0.375 | 0.221 | 0.293 | 0.302 | 0.323 | | |
| | 336 | 0.271 | 0.301 | 0.435 | 0.290 | 0.171 | 0.265 | 0.429 | 0.427 | 0.357 | 0.400 | 0.357 | 0.388 | 0.274 | 0.327 | 0.330 | 0.343 | | |
| | 720 | 0.338 | 0.348 | 0.464 | 0.307 | 0.210 | 0.297 | 0.440 | 0.453 | 0.394 | 0.436 | 0.433 | 0.422 | 0.368 | 0.384 | 0.378 | 0.368 | | |
| NLinear(1024) | 96 | 0.170 | 0.226 | 0.383 | 0.270 | 0.133 | 0.229 | 0.379 | 0.404 | 0.296 | 0.351 | 0.317 | 0.359 | 0.163 | 0.257 | 0.263 | 0.299 | 0.314 | 0.336 |
| | 192 | 0.215 | 0.265 | 0.397 | 0.274 | 0.148 | 0.242 | 0.414 | 0.426 | 0.337 | 0.382 | 0.352 | 0.381 | 0.216 | 0.294 | 0.297 | 0.323 | | |
| | 336 | 0.259 | 0.298 | 0.410 | 0.281 | 0.164 | 0.259 | 0.442 | 0.445 | 0.359 | 0.407 | 0.374 | 0.393 | 0.265 | 0.326 | 0.325 | 0.344 | | |
| | 720 | 0.321 | 0.342 | 0.449 | 0.303 | 0.203 | 0.292 | 0.470 | 0.477 | 0.470 | 0.456 | 0.409 | 0.413 | 0.338 | 0.375 | 0.372 | 0.379 | | |
| PIH(1024) | 96 | 0.147 | 0.198 | 0.357 | 0.248 | 0.127 | 0.220 | 0.360 | 0.394 | 0.263 | 0.328 | 0.291 | 0.349 | 0.161 | 0.253 | 0.244 | 0.284 | 0.297 | 0.326 |
| | 192 | 0.191 | 0.239 | 0.371 | 0.255 | 0.145 | 0.240 | 0.396 | 0.418 | 0.324 | 0.370 | 0.337 | 0.374 | 0.213 | 0.289 | 0.282 | 0.312 | | |
| | 336 | 0.241 | 0.280 | 0.392 | 0.261 | 0.160 | 0.256 | 0.409 | 0.432 | 0.314 | 0.376 | 0.360 | 0.386 | 0.265 | 0.326 | 0.306 | 0.331 | | |
| | 720 | 0.309 | 0.329 | 0.430 | 0.282 | 0.192 | 0.287 | 0.435 | 0.466 | 0.378 | 0.425 | 0.405 | 0.411 | 0.342 | 0.375 | 0.356 | 0.368 | | |

## A.4 PROOFS OF IB

### A.4.1 PROOF OF EQ. 8

We first examine the first term $-I\left(\mathbf{z}^{\text{noise}}, \mathbf{Y}\right)$ in Eq. 4 which encourages $\mathbf{z}_{\text{noise}}$ is informative of label $\mathbf{Y}$.

$$-I\left(\mathbf{z}^{\text{noise}}, \mathbf{Y}\right) \leq \mathbb{E}_{\mathbf{Y}, \mathbf{z}^{\text{noise}}} - \log q_\theta\left(\mathbf{Y} \mid \mathbf{z}^{\text{noise}}\right)$$
$$:= \mathcal{L}_{\text{pred}}\left(\mathbf{z}^{\text{noise}}, Y\right) \tag{13}$$

Here, $p_\theta\left(\mathbf{Y} \mid \mathbf{z}^{\text{noise}}\right)$ represents the variational approximation to the true posterior distribution $p\left(\mathbf{Y} \mid \mathbf{z}^{\text{noise}}\right)$ (a detailed proof can be found in Appendix A.4). This equation illustrates that minimizing $-I\left(\mathbf{z}^{\text{noise}}, \mathbf{Y}\right)$ is achieved by minimizing the prediction loss between $\mathbf{z}^{\text{noise}}$ and $\mathbf{Y}$. We choose to utilize the Mean Squared Error (MSE) loss to quantify the disparity between the prediction and the ground truth.

Here we provide more details about how to yield Eq. 13. By the definition of mutual information and introducing variational approximation $p_\theta\left(\mathbf{Y} \mid \mathbf{z}^{\text{noise}}\right)$ of intractable distribution $p\left(\mathbf{Y} \mid \mathbf{z}^{\text{noise}}\right)$, we have:

$$I\left(\mathbf{Y}, \mathbf{z}^{\text{noise}}\right) = \mathbb{E}_{\mathbf{Y}, \mathbf{z}^{\text{noise}}}\left[\log \frac{p\left(\mathbf{Y} \mid \mathbf{z}^{\text{noise}}\right)}{p(\mathbf{Y})}\right]$$
$$= \mathbb{E}_{\mathbf{Y}, \mathbf{z}^{\text{noise}}}\left[\log \frac{p_\theta\left(\mathbf{Y} \mid \mathbf{z}^{\text{noise}}\right)}{p(\mathbf{Y})}\right] \tag{14}$$
$$+ \mathbb{E}_{\mathbf{z}^{\text{noise}}}\left[KL\left(p\left(\mathbf{Y} \mid \mathbf{z}^{\text{noise}}\right) \| p_\theta\left(\mathbf{Y} \mid \mathbf{z}^{\text{noise}}\right)\right)\right]$$

According to the non-negativity of the KL divergence, we have:

$$I\left(\mathbf{Y}; \mathbf{z}^{\text{noise}}\right) \geq \mathbb{E}_{\mathbf{Y}, \mathbf{z}^{\text{noise}}}\left[\log \frac{p_\theta\left(\mathbf{Y} \mid \mathbf{z}^{\text{noise}}\right)}{p(\mathbf{Y})}\right]$$
$$= \mathbb{E}_{\mathbf{Y}, \mathbf{z}^{\text{noise}}}\left[\log p_\theta\left(\mathbf{Y} \mid \mathbf{z}^{\text{noise}}\right)\right] + H(\mathbf{Y})$$

We can ignore $H(\mathbf{Y})$ since it can be treated as a constant. We model $p_\theta\left(\mathbf{Y} \mid \mathbf{z}^{\text{noise}}\right)$ as a predictor parameterized by $\theta$, which generates the model prediction $\mathbf{Y}$ based on the input $\mathbf{z}^{\text{noise}}$. Thus, minimizing the upper bound of $-I\left(\mathbf{z}^{\text{noise}}, \mathbf{Y}\right)$ entails minimizing the model prediction loss $\mathcal{L}_{\text{pred}}\left(\mathbf{z}^{\text{noise}}, \mathbf{Y}\right)$. We opt to employ the Mean Squared Error (MSE) loss to quantify the difference between the prediction and the ground truth.

### A.4.2 PROOF OF EQ. 9

We derive the upper bound of $I\left(\mathbf{z}^{\text{noise}}, \mathbf{z}\right)$ by introducing the variation approximation $q\left(\mathbf{z}^{\text{noise}}\right)$ of distribution $p\left(\mathbf{z}^{\text{noise}}\right)$ :

$$
\begin{aligned}
I\left(\mathbf{z}^{\text{noise}}, \mathbf{z}\right) &= \mathbb{E}_{\mathbf{z}, \mathbf{z}^{\text{noise}}}\left[\log \frac{p_\phi\left(\mathbf{z}^{\text{noise}} \mid \mathbf{z}\right)}{p(\mathbf{z})}\right] \\
&= \mathbb{E}_{\mathbf{z}, \mathbf{z}^{\text{noise}}}\left[\log \frac{p_\phi\left(\mathbf{z}^{\text{noise}} \mid \mathbf{z}\right)}{q(\mathbf{z}^{\text{noise}})}\right] \\
&\quad - \mathbb{E}_{\mathbf{z}^{\text{noise}}, \mathbf{z}}\left[KL\left(p\left(\mathbf{z}^{\text{noise}}\right) \| q\left(\mathbf{z}^{\text{noise}}\right)\right)\right]
\end{aligned}
\tag{15}
$$

According to the non-negativity of KL divergence, we have:

$$
I\left(\mathbf{z}^{\text{noise}}, \mathbf{z}\right) \leq \mathbb{E}_{\mathbf{z}}\left[KL\left(p_\phi\left(\left(\mathbf{z}^{\text{noise}} \mid \mathbf{z}\right) \| q\left(\mathbf{z}^{\text{noise}}\right)\right)\right]
\tag{16}
$$

we assume that $q\left(\mathbf{z}^{\text{noise}}\right)$ is obtained by aggregating the patch representations in a fully perturbed sequences. The noise $\epsilon \sim \mathcal{N}\left(\mu_{\mathbf{z}}, \sigma_{\mathbf{z}}^2\right)$ is sampled from a Gaussian distribution where $\mu_{\mathbf{z}}$ and $\sigma_{\mathbf{z}}^2$ are mean and variance of $\mathbf{z}$. Choosing sum pooling as the aggregatiion function, since the summation of Gaussian distributions is a Gaussian, we have the following equation:

$$
q\left(\mathbf{z}^{\text{noise}}\right) = \mathcal{N}\left(N\mu_{\mathbf{z}}, N\sigma_{\mathbf{z}}^2\right)
\tag{17}
$$

Then for $p_\phi\left(\mathbf{z}^{\text{noise}} \mid \mathbf{z}\right)$, we have the following equation:

$$
p_\phi\left(\left(\mathbf{z}^{\text{noise}} \mid \mathbf{z}\right) = \mathcal{N}\left(N\mu_{\mathbf{z}} + \sum_{j=1}^{N} \lambda_j \mathbf{z}_j - \sum_{j=1}^{N} \lambda_j \mu_{\mathbf{z}}, \sum_{j=1}^{N}\left(1-\lambda_j\right)^2 \sigma_{\mathbf{z}}^2\right)
\tag{18}
$$

Finally, we have following inequality by plugging Equation 17 and Equation 18 into Equation Equation 16:

$$
I\left(\mathbf{z}^{\text{noise}}, \mathbf{z}\right) \leq \mathbb{E}_{\mathbf{z}}\left[-\frac{1}{2}\log A + \frac{1}{2N}A + \frac{1}{2N}B^2\right] + C
$$

where $A = \sum_{j=1}^{N}\left(1-\lambda_j\right)^2, B = \frac{\sum_{j=1}^{N} \lambda_j\left(\mathbf{z}_j - \mu_{\mathbf{z}}\right)}{\sigma_{\mathbf{H}^1}}$ and $C$ is a constant term which is ignored during optimization.

### A.5 EXPERIMENTS SETTINGS

PIH is built upon the PatchTST framework and thus incorporates all hyperparameters from PatchTST. To ensure a fair comparison, we adhered strictly to the settings of PatchTST for these shared hyperparameters, with the exception of the learning rate. We conducted a hyperparameter search only for those introduced by the HTM and IBF modules, as this was necessary. The only exception is the learning rate. Given the introduction of the Mamba and IBF modules, the default learning rate of $lr = 0.0001$ in PatchTST is suboptimal. Consequently, we set the search space for the PIH learning rate to $lr = \{0.001, 0.0005, 0.0001\}$. To ensure a fair comparison, we also performed a hyperparameter search for the learning rate in PatchTST, using $lr = \{0.001, 0.0005, 0.0001\}$, and selected the optimal results. The resulting mean Absolute Error (MAE) values were 0.310 and 0.335, which are almost unchanged compared to the default learning rate ($lr = 0.0001$), yielding 0.310 and 0.336. Thus, this does not affect our result analysis.

Our model incorporates several crucial hyperparameters, including $K$, which determines the number of partitions; $\beta$, which governs the balance between prediction and compression in the information bottleneck (IB) objective; and the temperature factor $\tau$, which influences subsequence sampling. We set $K \in \{2, 4\}$, $\beta \in \{0.0001, 0.001, 0.1, 1\}$, and $\tau \in \{0.1, 0.5, 1, 2\}$. We selected the optimal hyperparameters based on the results from the validation set.

Additionally, we analyzed the effects of these hyperparameters. The results indicate that the choice of $K$ does not significantly impact performance. In contrast, both $\tau$ and $\beta$ exhibit considerable influence on performance, likely due to variations in the redundancy levels across different datasets.

## A.6 MAMBA VS TRANSFORMER

We analyze our model from both performance and computational overhead perspectives and find that the hybrid architecture has distinct advantages over using only Mamba or Transformer.

**From a performance perspective**, the ablation experiments presented in Fig. 6 indicate that removing the Transformer results in slightly worse performance, highlighting the significant advantage of the combined Transformer and Mamba architecture. This finding is further supported by recent works such as Mamba-2-Hybrid (Waleffe et al., 2024), Dimba (Fei et al., 2024), and Jamba (Lieber et al., 2024).

**Considering computational overhead**, our framework employs the Transformer solely to process the partitioned short subsequences, which generally mitigates concerns about the costs associated with the Transformer. To validate this, we compared the computation time and GPU memory usage between using a single layer of Mamba and a single layer of Transformer under various lookback window settings (with nearly identical parameter counts). As shown in Fig. 4, when $L \leq 336$, the computational overhead of the Transformer is even lower than that of Mamba; however, at $L = 1024$, the computational cost of the Transformer is nearly twice that of Mamba. In our experiments, $K$ is typically set to 4, resulting in a subsequence length of $L/K = 1024/4 < 336$. Consequently, the addition of the Transformer module incurs less overhead compared to using only Mamba.

In summary, we conclude that retaining the Transformer module is essential for enhancing performance while managing computational costs effectively.

Table 4: Comparison of GPU memory usage and training time per epoch for a single-layer Transformer and Mamba on the Weather dataset as the lookback window $L$ varies.

|  |  | 96 | 192 | 336 | 512 | 1024 |
|---|---|---|---|---|---|---|
| **Mamba** | time (s) | 18.76 | 21.63 | 28.25 | 36.52 | 58.47 |
|  | memory (G) | 2.02 | 3.30 | 4.90 | 6.78 | 9.53 |
| **Transformer** | time (s) | 7.33 | 17.94 | 27.84 | 44.70 | 96.57 |
|  | memory (G) | 0.75 | 1.64 | 3.21 | 5.56 | 15.05 |

