# OpenReview forum: "Overcoming Lookback Window Limitations: Exploring Longer Windows in Long-Term Time Series Forecasting"
_ICLR.cc/2025/Conference — ICLR 2025 Conference Withdrawn Submission_

### Official Review · Reviewer_3NWY · 2024-10-29

**Soundness:** 1
**Presentation:** 2
**Contribution:** 1
**Rating:** 3
**Confidence:** 3

**Summary:**

This paper addresses the look-back Window Limitation problem in Long-term Time Series Forecasting (LTSF). Current Transformer-based models suffer from the look-back window limitation, and as the look-back window increases, computational complexity grows and may include redundant information that interferes with learning. The paper proposes two modules: the Information-Bottleneck-Filter module, which improves performance by preventing information redundancy in LTSF, and the Hybrid-Transformer-Mamba (HTM) module, which enhances long-range modeling performance by applying Mamba to patched sequences. The model created by combining these two modules with the PatchTST model showed good performance in LTSF tasks with a long look-back window of 1024.

**Strengths:**

* Demonstrates the necessity of models with longer look-back windows and shows the limitations of existing transformers.
* Conducted experiments using input sequences of length up to 1024.
* Combined the Mamba-Transformer hybrid with the information bottleneck approach.

**Weaknesses:**

* The overall completeness is significantly lacking.
  - There is no explanation for Figure 2 in both the caption and main text, and the caption was written carelessly.
  - While Lookback Window Limitation is a key concept appearing from the title, there is no definition or explanation for it, and the cited papers do not address this concept either.
  - Revise the artifact in Figure 3(a).
  - Add explanations for each axis in the graphs (Figure 3,5), and when abbreviations are used, write out the full terms in the caption (Figure 6).
  - Instead of mixing the terms Ettm and ETTm, standardize the usage to ETTm.

* Since this paper's main component is the Mamba-Transformer hybrid, it should: 1. Include recently proposed Mamba-Transformer hybrid models in the related work section 2. Compare performance with these models in the main table. e.g. *SST: Multi-Scale Hybrid Mamba-Transformer Experts for Long-Short Range Time Series Forecasting Can Mamba Learn How to Learn?*, *A Comparative Study on In-Context Learning Tasks*

* Please revise the disorganized notations and the inappropriate figures in the Methods section, and include sufficient explanations about the model components, even if in the appendix.

* There is no explanation about how many seeds were used for repeated experiments. The outlier values raise suspicions that no repeated experiments were conducted at all, reducing the reliability of the results.

* In Table 1, PIH's performance appears to largely come from PatchTST's excellence. Please provide a full table that compares basemodel and basemodel+IBF+HTM, which is briefly introduced in section 4.2.

* Recent time series forecasting papers generally use more public datasets. Please add datasets such as ECL, Exchange, Solar-Energy, and PEMS.

* Looking at Figure 5, your methodology doesn't appear to effectively filter redundant information or efficiently utilize longer inputs as the look-back period increases. More experiments on additional datasets and sufficient explanations of the results seem necessary.

* Is your methodology also applicable to the iTransformer model?

While this paper attempts to tackle an important problem in time-series forecasting and has some interesting aspects, in my opinion, it lacks distinctiveness compared to existing transformer-mamba hybrids, and the paper shows insufficient overall completeness and ability to demonstrate its results.

**Questions:**

See Weaknesses

---

### Official Review · Reviewer_gzbk · 2024-10-29

**Soundness:** 2
**Presentation:** 3
**Contribution:** 2
**Rating:** 3
**Confidence:** 4

**Summary:**

This paper addresses the challenges of long-term time series forecasting (LTSF) by tackling the Lookback Window Limitation (LWL) encountered in current Transformer-based models. The authors argue that while longer lookback windows can theoretically provide richer historical insights, they often lead to redundancy and overfitting to temporal noise, particularly given the quadratic complexity of Transformers.

To mitigate these issues, the paper introduces the Information Bottleneck Filter (IBF), which employs information bottleneck theory to extract crucial subsequences from input data, thereby reducing redundancy. Additionally, the authors propose the Hybrid-Transformer-Mamba (HTM), a novel architecture that combines the linear complexity and long-range modeling capabilities of Mamba with the short-sequence strengths of Transformers.

Through integration of these model-agnostic components into existing methods, the authors conduct experiments across seven datasets, demonstrating significant improvements in performance. Notably, the introduction of the PIH (Patch-IBF-HTM) strategy enables the extension of the lookback window to 1024, achieving state-of-the-art results and underscoring the value of utilizing longer historical sequences for forecasting tasks.

**Strengths:**

The paper introduces the Information Bottleneck Filter (IBF) and Hybrid-Transformer-Mamba (HTM) as novel approaches to address the Lookback Window Limitation (LWL) in long-term time series forecasting. The integration of these concepts demonstrates an attempt to enhance model performance and reduce redundancy in input data. The authors conduct experiments across seven datasets, providing empirical validation of their methods, which achieve competitive results in the context of existing literature. The paper is organized clearly, making it accessible to readers.

**Weaknesses:**

1. **Concerns on SOTA Comparisons**: As a submission for 2025, it is surprising that the authors only compare with the 2023 PatchTST and the 2024 S-Mamba. Additionally, the comparisons with FEDformer and Autoformer (2022 and 2021, respectively) do not adequately represent the current state of the art. Please annotate each method with its publication year to provide context. While including S-Mamba is valuable, it is still an unpeer-reviewed work. A more comprehensive comparison with recent methods would enhance the submission's relevance and credibility.

2. **Fairness of Window Size Comparisons**: The parameter settings for the baselines in Table 1 are not clearly defined. Results for FEDformer, Autoformer, and Informer seem to replicate those from PatchTST, but their lookback window $ L $ does not uniformly match the PIH setting of 1024. This raises fairness concerns and contradicts the authors' claim of not intentionally disadvantaging these models. I recommend creating two tables: one where all models are evaluated at $ L = 1024 $ and another using each model’s optimal $ L $. Presenting both together would effectively demonstrate the significance of PIH's larger lookback window.

3. **Experimental Rigor**: The authors state that PIH outperforms other models under shorter windows (e.g., 336 and 512), but this claim lacks rigor since only DLinear and NLinear are compared. According to Table 1, PatchTST appears to show superior performance. Additionally, since both PatchTST and DLinear/NLinear assume channel independence, it is crucial to compare with channel-dependent models in this context.

4. **Minor Corrections**:
   - In Figure 6, the left panel legend incorrectly states "PHI" instead of "PIH."
   - There is a duplicate paragraph on page 2: "Moreover, it does not mitigate the quadratic complexity inherent in Transformers." This should be corrected.

**Questions:**

I find it puzzling that while the authors identify the Lookback Window Limitation (LWL) and propose corresponding solutions, they also state in the conclusion that "longer lookback windows are not always beneficial for all datasets." This suggests that the core issue may not be fully addressed. Could the authors provide a more detailed explanation of this apparent contradiction?

---

### Official Review · Reviewer_RXSX · 2024-11-01

**Soundness:** 3
**Presentation:** 3
**Contribution:** 2
**Rating:** 6
**Confidence:** 4

**Summary:**

The paper addresses the Lookback Window Limitation (LWL) in long-term time series forecasting, where current models struggle with longer sequences due to redundancy and overfitting. To solve this, the authors propose the Information Bottleneck Filter (IBF) to reduce redundancy and the Hybrid-Transformer-Mamba (HTM) model to handle long sequences efficiently. These solutions enable improved performance with extended lookback windows.

**Strengths:**

The paper introduces a novel approach to overcoming the Lookback Window Limitation (LWL) in long-term time series forecasting, which is a significant research contribution. The paper innovates both theoretically and practically by introducing the Information Bottleneck Filter (IBF) and the Hybrid-Transformer-Mamba (HTM) model.

**Weaknesses:**

1. The paper mentions the use of HYBRID-TRANSFORMER-MAMBA (HTM) to model long and short time series information separately, but it does not provide diagrams to detail how the Transformer and Mamba models are combined, reducing the comprehensibility of the method.
2. Table 1 only shows results for L=1024, but some baseline methods may perform better with shorter lookback windows; therefore, it would be beneficial to provide the performance of each method under its original configuration.
3. The authors select the best results for T=720 by choosing different lookback windows {24, 48, 96, 192, 336, 720, 1024}, which may seem unfair. The theory proposed by the authors should not only apply to long sequences; thus, the model's ability to extract information for any input length should be superior to the current best methods.
4. The visualization results in Figure 3 (c) are intriguing. Could the authors please explain how this figure was obtained?

**Questions:**

1. Can a comparison of other baselines with different lookback window lengths?
2. Why was the choice of 1024 made, and how does the performance with longer window lengths? Could more experimental analysis be provided to demonstrate that the model can indeed extract information more relevant to prediction?
3. The paper mentions several key hyperparameters, such as K, β, and τ. Does the selection of these parameters have a significant impact on model performance? Why is there no discussion on the choice of parameters in the experimental results?

---

### Official Review · Reviewer_WAXz · 2024-11-04

**Soundness:** 1
**Presentation:** 2
**Contribution:** 1
**Rating:** 1
**Confidence:** 5

**Summary:**

The paper explores the topic of long-horizon forecasting using transformer models. The key contribution of the paper is in the sequence sampling method relying on a mixture of Mamba and information bottleneck filtering technique  for patch selection to increase the input window size to 1024.

**Strengths:**

- The paper focuses on increasing the input window dimensionality, which is also an active research are in LLM research

**Weaknesses:**

- The long-term datasets are overused in the literature, perhaps also implying overfitting after so many modeling iterations have been made over these datasets. There is no evidence that any meaningful relations exist between variables in these datasets to support multivariate forecasting. Datasets are small and do not reflect the complexities that exist in production grade big datasets employed in industrial environment. As a result, we are most likely looking at overfitting artifacts when people continue perusing these datasets. I do not see evidence that there is statistical significance in results that researchers in the field continue to squeeze from these degenerate datasets.  The field should move on resolving more interesting and realistic large scale forecasting problems.

- How do the multivariate results presented in the paper compare against univariate baselines. I do not believe there is evidence that these multi-variate models can even beat the univariate baselines, making the entire set of results look unconvincing to me. Please provide comparison between multivariate forecasting result and univariate forecasting results on the same data. I expect that the multivariate forecasting result should be significantly better than univariate to justify the additional complexity.

- There is no evidence in the paper that setting lookback length to 1024 creates problems in other architectures. There is no evidence in the paper that increasing the length of the input to 1024 leads to significant accuracy lift in the proposed architecture. This basically undermines the pitch of the title "Overcoming Lookback Window Limitations...". Are there really significant limitations and the introduction of the current technique really solves it? Please provide evidence that increasing the lookback length to 1024 in the proposed architecture significantly improves accuracy compared to, say, lengths 512, 256 or 128.

- The key focus of the paper is comparison against PatchTST. This is a 2-year old model with 900+ citations. Comparison against recent state of the art is missing. For example, https://proceedings.mlr.press/v238/zhang24l/zhang24l.pdf, https://www.ijcai.org/proceedings/2024/0608.pdf, https://arxiv.org/pdf/2405.14616. This is a good reference for recent SOTA models https://arxiv.org/pdf/2403.09898

- The application of the proposed technique to the most recent models is missing, making the results unconvincing. The authors claim that they developed a general technique. In all key studies they apply it only to PatchTST. In ablation study Figure 4 they apply only to Transformer (I understand this is vanilla transformer), Informer and Autoformer. All the the three models are so bad, I think doing anything on top of them, like a residual linear layer or MLP in parallel from input to output should do the job of improving their results and making them **at least** as good as the DLinear or NLinear. This intuition is supported by the fact that on the dataset used for the ablation, ETTm1, all these models are not even close to the linear baselines and they do not get much closer after the proposed technique is applied. Therefore, the results in this area are weak, making the claim ill-supported. Please provide evidence that this approach provides lift on more recent strong transformer-based architectures other than PatchTST.

**Questions:**

See weaknesses section

---

### Note · Authors · 2024-11-25

I have read and agree with the venue's withdrawal policy on behalf of myself and my co-authors.